# A multi-model approach to monitor emissions of CO<sub>2</sub> and CO in an urban-industrial complex

Ingrid Super<sup>1,2</sup>, Hugo A.C. Denier van der Gon<sup>2</sup>, Michiel K. van der Molen<sup>1</sup>, Hendrika A.M. Sterk<sup>3</sup>, Arjan Hensen<sup>4</sup>, Wouter Peters<sup>1,5</sup>

<sup>1</sup> Meteorology and Air Quality Group, Wageningen University, P.O. Box 47, 6700 AA Wageningen, Netherlands <sup>2</sup> Department of Climate, Air and Sustainability, TNO, P.O. Box 80015, 3508 TA Utrecht, Netherlands

<sup>3</sup> National Institute for Public Health and the Environment, P.O. Box 1, 3720 BA Bilthoven, Netherlands

<sup>4</sup> Energy Research Centre of the Netherlands, P.O. Box 1, 1755 ZG Petten, Netherlands

<sup>5</sup> Centre for Isotope Research, Energy and Sustainability Research Institute Groningen, University of Groningen,
 Nijenborgh 4, 9747 AG Groningen, Netherlands

Correspondence to: Ingrid Super (ingrid.super@wur.nl)

**Abstract.** Monitoring urban-industrial emissions is often challenging, because observations are scarce and regional atmospheric transport models are too coarse to represent the high spatiotemporal variability in the resulting concentrations. In this paper we present a new combination of a Eulerian model (WRF-Chem with an

- urban parameterisation) and a Lagrangian transport-deposition model (OPS), demonstrating that a plume model strongly improves our ability to capture urban plume transport. This follows from a comparison to observed CO<sub>2</sub> and CO mole fractions at four sites along a transect from an urban-industrial complex (Rotterdam, Netherlands) towards rural conditions. At the urban measurement site we find strong enhancements of up to 33.1 ppm CO<sub>2</sub> and 84 ppb CO over the rural background concentrations. These signals are highly variable due to the presence
- of distinct source areas dominated by road traffic/residential heating emissions or industrial activities. This causes different emission signatures that are observed in the CO:CO<sub>2</sub> ratios and can be well-reproduced with our framework, suggesting that top-down emission monitoring within this urban-industrial complex is feasible. Further downwind from the city, the urban plume is less frequently observed and its concentration becomes smaller and less variable, making these locations more suited for an integrated emission estimate over the whole
- study area. We find that WRF-Chem, although able to represent mesoscale patterns, lacks spatiotemporal detail to reproduce the timing, magnitude and variability of urban plumes at the regional background and urban sites. The implementation of the OPS plume model improves the simulation of the  $CO_2$  and CO enhancements. The bias for extreme  $CO_2$  pollution events is reduced with almost 80% from 15.4 to 3.4 ppm, while the reproducible fraction of observed variability over 750 measurements more than doubles (to 38%) with the use of a plume
- model. Therefore, we argue that a plume model with detailed and accurate dispersion parameters is crucial for top-down monitoring of greenhouse gas emissions in urban environments. Future research could benefit from assimilating observed wind fields to improve the plume representation at urban scales.

# **1** Introduction

Cities are major contributors to anthropogenic  $CO_2$  and air pollutant emissions (Brioude et al., 2013; Turnbull et 35 al., 2015; Velasco et al., 2014). Monitoring of urban emissions has therefore received a lot of attention (Font et al., 2014; Mays et al., 2009; Rayner et al., 2014; Silva et al., 2013; Wunch et al., 2009). In addition, several

studies have used high-resolution atmospheric transport models to simulate urban concentrations of  $CO_2$  and coemitted air pollutants, such as CO and  $NO_x$ , over megacities and larger regions (Brioude et al., 2013; Huszar et al., 2016; Lac et al., 2013; McKain et al., 2012; Ribeiro et al., 2016; Tolk et al., 2009; Zhang et al., 2015).

- However, modelling urban atmospheric composition remains challenging as the urban environment is rather complex and current emission inventories contain substantial uncertainties (Pouliot et al., 2012; Vogel et al., 2013). This implies that inversion studies that use a combination of models and observations to constrain fluxes have to deal with additional uncertainties caused by models. As such, inversion studies benefit from a detailed and accurate model that allows the model-observation mismatch to be attributed to errors in the emission
- inventory rather than transport or aggregation errors (Boon et al., 2016). Previous inversion studies therefore rely heavily on a strict data selection to favour well-mixed conditions with more reliable model output, which results in very small data sets and therefore increased uncertainty (Bréon et al., 2015; Brioude et al., 2013). Improving the model representation of urban transport is thus an essential step.
- In this paper we address two important concerns in the context of inverse modelling. The first issue is what 50 type of measurement location (urban vs. rural) can provide sufficient information on urban fluxes. Generally, urban sites are most exposed to local fluxes and therefore show a large variability (Bréon et al., 2015; Lac et al., 2013). In contrast, rural sites barely show a response to urban emissions due to the small range of wind direction at which the site is affected by the urban area. Moreover, the dilution of urban plumes increases with distance (Calabrese, 1990; Finn et al., 2007) and the observed signal at the rural site can be small. A final consideration is
- that near-ground measurements, as commonly found in cities, are highly influenced by local sources that dominate the overall urban signal. Boon et al. (2016) suggested that, even if strict data selection is applied, the use of such sites in inversions with high-resolution Eulerian models might be limited. Together, these papers suggest that a useful measurement location should be just downwind of an urban area relative to the dominant wind direction at a distance that ensures enough exposure to the urban plume and limits model errors due to large
- heterogeneity and local emissions. We will examine our transect of measurement sites to see which site best matches this criterion.

The second question we address is what type of modelling framework is best capable of explaining urban plume transport and the resulting mole fractions at the measurement sites. In atmospheric composition modelling both Eulerian and Lagrangian (plume or puff) models are used, or a combination of both (Kim et al., 2014;

- Korsakissok and Mallet, 2010b). Eulerian models use a grid that can be adapted to cover either small or large areas in different resolutions and are therefore widely used. However, Eulerian models instantly mix chemical species and their emissions through the grid box, which enhances urban plume dispersion in the horizontal and vertical. The resulting errors in the transport and mixing are reflected in unrealistic concentrations (Karamchandani et al., 2011; Tolk et al., 2009). How large the corresponding error is depends on the
- heterogeneity of the emissions and the grid resolution (Tolk et al., 2008). A plume model improves the description of horizontal and vertical mixing and can account for higher spatial heterogeneity of emissions and concentrations. The use of such models has proven useful for both inert (Rissman et al., 2013) and chemically reactive species (Briant and Seigneur, 2013; Korsakissok and Mallet, 2010b; Vinken et al., 2011). However, a plume model is computationally demanding when used over a large area with many sources.
- Therefore, we compare three modelling frameworks which consist of the Eulerian WRF-Chem model and the Lagrangian OPS plume model forced by different meteorological fields. The model output is compared to long-

term observations of  $CO_2$  and CO at several measurement sites along an urban-to-rural transect. Oney et al. (2015) have previously examined an extensive  $CO_2$ ,  $CH_4$  and CO measurement network in combination with the FLEXPART-COSMO model. However, their framework focused on regional (~100-500 km), terrestrial fluxes

- and contained no urban observation sites nor a model inter-comparison. To our knowledge, this is the first study to combine three modelling frameworks for  $CO_2$  and CO in an urban setting with four distinct measurement sites. Since the measurement location determines the level of spatiotemporal variation that can be observed in the plume, it also determines the requirements imposed on the modelling framework. Thus with this study set-up we examine whether a Eulerian model can represent the urban plume signals at the measurement sites and under
- which conditions a plume model improves this representation. We use the Rijnmond area (The Netherlands) including the city of Rotterdam as our case study, which is surrounded by scattered urban, agricultural, and rural areas. We chose this area because of the availability of a 1x1 km<sup>2</sup> emission inventory and its complex combination of urban and industrial activities.
- This paper starts with a description of the modelling frameworks (Sect. 2.1 and 2.2), observations, and study 90 design (Sect. 2.3). Subsequently, we examine the ability of WRF-Chem to represent mesoscale changes in the  $CO_2$  and CO concentrations (Sect. 3.1). WRF-Chem will also determine the large-scale background of the Lagrangian simulations and a good representation of this background is necessary before looking a finer scales. We then discuss the four measurement sites and their capability to detect urban signals, and demonstrate the added value of both urban and regional background sites (Sect. 3.2). Section 3.3 examines the ability of WRF-
- Chem to represent the urban signals at the measurement sites. We do this by selecting specific events when the wind advects pollutants from the Rijnmond area. Finally, we discuss the advances made by implementing a plume model based on the same events and examine under which conditions a plume model improves the WRF-Chem plume representation. All these results lead to recommendations for future monitoring and modelling of urban atmospheric composition in Sect. 4.

## 100 2 Methods

#### 2.1 WRF-Chem

The Eulerian model used in this study is WRF-Chem V3.2.1 (Skamarock et al., 2008). To simulate concentration fields of  $CO_2$  and CO we implement their budgets. These are described below, followed by a description of the model set-up.

## 105 2.1.1 The CO<sub>2</sub> budget

The atmospheric  $CO_2$  mole fraction at a particular location is a function of several terms. Here we follow the methodology used by Bozhinova et al. (2014), described in Eq. (1).

$$X_{CO2,obs} = X_{CO2,bg} + X_{CO2,ff} + X_{CO2,p} + X_{CO2,r} + X_{CO2,o} + X_{CO2,s}$$
(1)

where the indices express the origin of CO<sub>2</sub>: obs – total observed concentration at a particular location, bg –
 background mole fraction, ff – fossil fuels, p – photosynthetic uptake, r – ecosystem respiration, o – ocean, s –
 stratosphere-troposphere exchange. In order to simplify Eq. (1), we suggest to omit some of these terms.

Several studies show that the stratosphere-troposphere exchange is an important process in determining the concentration of chemical species in the troposphere (Esler et al., 2001; Holton et al., 1995). Shia et al. (2006)

find that the concentration and temporal evolution of CO<sub>2</sub> in the mid-latitudes are affected by the Brewer-115 Dobson circulation, which transports stratospheric air to the troposphere. However, their results suggest that it takes about three years to complete one cycle from troposphere to stratosphere and back again. In addition, Sportisse (2010) suggests a characteristic timescale for exchange from the stratosphere to the troposphere of 1–2 years. Given we have simulations of only three months, we consider that the effects of this circulation on the CO<sub>2</sub> mole fraction are captured in the background mole fraction and any changes in the circulation are negligible

within the timeframe of this study.

Similarly, the influence of global ocean fluxes is assumed to be small, with only the nearby Northern Atlantic ocean representing substantial uptake (~ $0.25 \text{ Pg C yr}^1$ ) near our domain (Lee et al., 2003; Mikaloff Fletcher et al., 2006). This uptake is well-captured in the CarbonTracker global CO<sub>2</sub> system that provides the boundary conditions for our study, and we assume the influence of oceans and coastal seas within our domain to be negligible.

By omitting the stratosphere and ocean terms, Eq. (1) can be simplified and the change in CO<sub>2</sub> mole fraction at a particular location between two time steps can be calculated using Eq. (2).

$$\Delta X_{CO2,obs} = \Delta X_{CO2,bg} + E_{CO2,ff} - A_n + R \tag{2}$$

where  $E_{CO2,ff}$  is the fossil fuel emission,  $A_n$  is the net primary productivity by vegetation and R is the 130 heterotrophic ecosystem respiration. The change in background CO<sub>2</sub> mole fraction ( $\Delta X_{CO2,bg}$ ) is caused by advection.

#### 2.1.2 The CO budget

The main sources of CO are fossil fuel combustion and oxidation of hydrocarbons (US EPA, 1991). Gerbig et al. (2003) argued that the oxidation term is important for the background CO concentration. Yet, they found that the CO fraction from local anthropogenic emissions dominates at the measurement sites. We assume this will also be the case in the urban-industrial environment of our case study. The main sink of CO is the reaction with the hydroxyl radical (chemical loss term  $L_{CO}$ ), which we account for with a simple decay function. We assume steady-state, i.e. the OH concentration is taken relatively small (10<sup>6</sup> molecules cm<sup>-3</sup>) and constant. This results in a lifetime for CO of about 2 months at mid-latitudes (Jacob, 1999):

$$\Delta X_{CO,obs} = \Delta X_{CO,bg} + E_{CO,ff} + L_{CO} \tag{3}$$

#### 2.1.3 Model set-up

We use meteorological fields from the National Centers for Environmental Prediction (NCEP) Final (FNL)
Operational Global Analysis (National Centers for Environmental Prediction/National Weather Service/NOAA/U.S. Department of Commerce, 2000) at 1x1° horizontal resolution and a temporal resolution of
6 hours. We define four 2-way nested domains (Fig. 1) which have a horizontal resolution of 48x48, 12x12, 4x4 and 1x1 km respectively, and a vertical resolution of 29 eta levels. The outer domain is situated over Europe. Domains 2–4 zoom in on the Rijnmond area in the southwest of the Netherlands. We have used the Yonsei University (YSU) boundary layer scheme (Hong et al., 2006), the Dudhia scheme for shortwave radiation (Dudhia, 1989), the Rapid Radiation Transfer Model (RRTM) as longwave radiation scheme (Mlawer et al., 1997), and the Unified Noah Land-Surface Model as the surface physics scheme (Ek et al., 2003). We also used

the single-layer urban canopy model (UCM) to account for changes in the roughness length and heat fluxes in the urban environment (Chen et al., 2011).

Separate tracers are implemented that describe the different contributions (Eq. (2) and Eq. (3)) to the total concentrations of CO<sub>2</sub> and CO. The background CO<sub>2</sub> is purely determined by the initial and boundary

- conditions, which are taken from the 3D mole fractions from CarbonTracker (Peters et al., 2010). The CarbonTracker 3D fields have a horizontal resolution of 1x1° and 34 vertical levels. Therefore, it is both horizontally and vertically interpolated onto the WRF-Chem grid. The CO initial and boundary conditions are calculated with IFS-MOZART (Flemming et al., 2009) from the Monitoring Atmospheric Composition and Climate (MACC) project. The boundary conditions are updated every 6 hours (only for the outer domain).
- The fossil fuel emissions for domains 1–3 are taken from the TNO-MACC III inventory for 2011 (Kuenen et al., 2014) and have a horizontal resolution of 0.125x0.0625°. All fossil fuel emissions for domain 4 are collected from the Dutch Emission Registration (Netherlands PRTR, 2014) and compiled by TNO (Netherlands Organization for Applied Scientific Research) for the year 2012. The emissions are divided over ten SNAP emission categories, summarised in Table 1. We apply a temporal profile to the emissions by assigning hourly,
- daily and monthly fractions to the emissions per emission category (Denier van der Gon et al., 2011). Area source emissions are added to the lowest surface model level every hour. Point source emissions (only SNAP 1, 3, 4, 8 and 9) are given a simplified, fixed vertical distribution based on previous research with plume rise calculations (Bieser et al., 2011), also shown in Table 1.
- The biogenic  $CO_2$  fluxes are generated in the same way as done by Bozhinova et al. (2014). With the SiBCASA model (Schaefer et al., 2008) monthly averaged  $1x1^{\circ} A_n$  and *R* are calculated for nine different land use types. Combining the high-resolution land-use map of WRF with the SiBCASA fluxes gives us biogenic fluxes on the resolution of the WRF grid. The temporal resolution is enhanced by scaling the  $A_n$  and *R* at each WRF-Chem time step with modelled shortwave solar radiation and 2m temperature:

 $A_n = A_{n,f} \cdot SW_{in}$ (4)  $R = R_f \cdot 1.5^{(T_2 - 273.15)/10}$ (5)

where  $A_{n,f}$  is the monthly average photosynthetic flux (mole CO<sub>2</sub> km<sup>-2</sup> h<sup>-1</sup>),  $SW_{in}$  the incoming shortwave solar radiation (W m<sup>-2</sup>),  $R_f$  the monthly average respiration flux (mole CO<sub>2</sub> km<sup>-2</sup> h<sup>-1</sup>), and  $T_2$  the 2m temperature (K). This procedure was first described in Olsen and Randerson (2004).

#### 2.2 The plume dispersion model OPS

- The plume dispersion model OPS (Operational Priority Substances) is a Gaussian plume model that calculates the transport, dispersion, chemical conversion and deposition of pollutants (Sauter et al., 2016; Van der Swaluw et al., 2011; Van Jaarsveld, 2004; Van Jaarsveld and Klimov, 2011). It is used to calculate large-scale, yearly averaged concentration and deposition maps for the Netherlands at 1x1 km<sup>2</sup> resolution. It was initially developed by the National Institute for Public Health and the Environment (RIVM) to model dispersion of pollutants like
- particulate matter and ammonia, but has also been used to study the dispersion of pathogens (Van Leuken et al., 2015). In this paper we use the short-term version of this model (version 10.3.5) to calculate the transport of point and/or area source emissions towards receptor points. The short-term model provides hourly concentrations at receptors that can be either site specific or gridded. In this study, only the contributions of local emissions to the concentration at the receptor sites are taken into account. The large-scale background concentrations are

taken from the WRF-Chem simulation and are added to the OPS plume concentrations to obtain the total concentration.

The OPS model is intended to use primary meteorological conditions as observed by the Dutch meteorological institute, and calculates secondary variables such as boundary layer height and friction velocity. We used the OPS model accordingly. Since this could result in an unfair comparison of WRF-Chem and OPS, we also

- replaced the primary parameters and OPS-calculated boundary layer height with those calculated by WRF-Chem in an additional OPS simulation. Although the original meteorological values are often a weighted average of multiple observations rather than a point observation, replacing them with point observations close to the urban Zweth site (see study design) has no significant impact on our findings. Therefore, we argue that using model output at Zweth allows a valid comparison between both models. Note that the meteorological conditions in OPS
- remain constant during each hour.

The OPS model was adapted to calculate dispersion of CO and  $CO_2$ . For both CO and  $CO_2$  we assume wet deposition plays no role due to their relative insolubility. Dry deposition of  $CO_2$  and CO is not included in the OPS model. Instead, the dry deposition of  $CO_2$ , i.e. biogenic uptake, is accounted for by WRF-Chem, while the dry deposition of CO is neglected. We assigned temporal profiles to the emissions dependent on the source type similar to the ones used in the WRF-Chem simulations.

### 2.3 Study design

We take the Rijnmond area as case study in which Rotterdam is the major urban area. The area is situated near the coast in the west of The Netherlands (inner domain in Fig. 1, also see Fig. 2) and includes a large harbour and industrial area. The terrain is flat. We have installed two measurement sites for observing CO<sub>2</sub> and CO 15
km south (Westmaas, 51.79° N, 4.45° E) and 7 km northwest (Zweth, 51.96° N, 4.39° E) of the city with an inlet at 10 m. We consider Zweth to be an urban site which is highly affected by urban fluxes. Westmaas functions as a background site as it is usually located upwind of the major source areas. These measurements have been

described in more detail by Super et al. (2016).

In addition to these local stations we include two regional background sites in our framework (see Fig. 1). The

- Cabauw site (51.97° N, 4.93° E) is situated 32 km east of the centre of Rotterdam. At this location CO<sub>2</sub> is measured at several heights (20, 60, 120 and 200 m) along a 200 m tall tower by the Energy research Centre of the Netherlands (ECN). CO is measured at ground level (2.5–4 m) by the RIVM. Another observation site is located at Lutjewad (53.40° N, 6.35° E), close to the coast in the north of the Netherlands. Here CO and CO<sub>2</sub> concentrations are observed at 60 m height (Van der Laan et al., 2009). These stations provide a transect from the city towards rural areas.
  - For the Cabauw  $CO_2$  measurements we selected the 60 m level. On average the  $CO_2$  concentrations are similar at all levels during well-mixed daytime conditions (Vermeulen et al., 2011). However, a large gradient can be observed for stable conditions or during night time when the 20 m level is highly affected by local fluxes. Similarly, Turnbull et al. (2015) suggested that measurements closer to the surface are more sensitive to local
- fluxes. Since we are interested in the plume from Rotterdam and not fluxes close to Cabauw, observations at a higher level than 20 m were more suitable. We choose the 60 m level observations to be able to compare easily to the Lutjewad site. However, a higher level could also be used without affecting our conclusions.

We simulated a period of 3 months, October–December 2014. We choose this period because of the high data coverage at all measurement sites. We did three types of simulations, using different models for calculating the plume concentration and using different meteorological input (Table 2). The WRF-Chem simulation provides the large-scale CO<sub>2</sub> and CO concentrations that are used as background in addition to the OPS simulations of the plume concentration. The OPS simulation includes all area and point sources in the Rijnmond area (similar to domain 4 of WRF-Chem). In addition, we gave the emissions from the inner domain in WRF-Chem different labels to be able to separate the urban plume from other signals. To identify the importance of a correct representation of meteorological conditions (wind speed, wind direction, temperature and humidity) we do the OPS simulations with both WRF-Chem meteorology and observed, interpolated meteorology (affix '\_wm or '\_om').

An advantage of OPS compared to WRF-Chem is that more detailed point source characteristics can be included. Therefore, we compared two OPS simulations; a 'simplified' run that includes the vertical distribution used in WRF-Chem (Table 1) and a 'detailed characteristics' run that includes detailed stack height and heat content information for plume rise calculations. At the Zweth observation site the CO<sub>2</sub> concentration resulting from point source emissions is on average 2.3 ppm smaller when the full characteristics are included, lowering the bias from 3.1 to 0.8 ppm. Therefore, all OPS simulations in this study are done with the full source characteristics.

#### 245 3 Results

#### 3.1 Background concentrations in WRF-Chem

Partitioning of total concentrations into different sources and sinks can be very useful to identify dominant contributors and their characteristics. Whereas observations are difficult to partition, our WRF-Chem simulations contain separate tracers for all sources/sinks. Figure 3 shows how the CO<sub>2</sub> and CO concentrations at the Zweth
location are build up according to the WRF-Chem simulation, separating between the background (i.e. advected from the model's boundary conditions), biogenic sources/sinks, and fossil fuel fluxes. Observed total concentrations are given by black dots. Whereas the CO signal can be dominated by fossil fuel fluxes, the CO<sub>2</sub> signal mostly consists of a background. Note that the y-axis starts at 350 ppm.

The fossil fuel fluxes can be separated into local and regional fluxes. The regional contribution comes from anywhere outside the Rijnmond area and is therefore dependent on wind direction. A change in wind regime will cause another area to be sampled, which generally means that an easterly wind will result in a larger contribution of (regional) fossil fuel fluxes (e.g. during the second half of November). This can be considered a mesoscale change. Hence, for the purpose of this study we will consider the regional fossil fuel fluxes to be part of the WRF-Chem background. Similarly, the biogenic contribution is strongly correlated with wind direction. With an

- easterly wind the advected air has a continental origin and this results in a larger biogenic contribution. In contrast, the local fossil fuel signal comes from sources in the direct vicinity of the measurement sites. Due to the short travel time it is less diluted and more variable than the regional fossil fuel contribution. In this paper we are mostly interested in the local fossil fuel signal, but Fig. 3 illustrates the importance of the other contributions. So before we focus on the local fossil fuel signal we first need to examine the ability of WRF-Chem to properly
- simulate the mesoscale changes in the background and biogenic signals.

The time series of the  $CO_2$  and CO concentrations at the Zweth and Cabauw measurement sites are shown in Fig. 4. The concentrations respond to a change in meteorological regime during the second half of November when the wind becomes easterly. As explained before, easterly winds advect a significant amount of regional fossil fuel  $CO_2$  and CO, resulting in higher observed concentrations. These large-scale patterns in the

- concentrations are captured well by WRF-Chem. Statistics for daily mean concentrations of the full simulation period (Table 3) show that the  $CO_2$  time series are simulated reasonably well and biases are small compared to the average observed  $CO_2$  concentrations. These results indicate that WRF-Chem is indeed able to reproduce mesoscale changes over time. For CO we find slightly smaller correlations except at Zweth, but the biases are relatively much larger compared to the average observed CO concentrations. Whereas the  $CO_2$  concentration has
- a substantial background contribution, the CO concentration is relatively more influenced by fossil fuel combustion which is more variable and therefore more challenging to simulate. However, during the period with easterly winds the model seems to underestimate the  $CO_2$  and CO concentrations and removing November from the analysis generally leads to better correlations and smaller biases for CO (see values between brackets in Table 3). Because the wind sector in which our measurement sites detect the Rijnmond plumes is located in the
- west, these results give confidence that our background concentrations will be simulated correctly. Correlations using afternoon (12-16h) average concentrations (not shown) are similar to daily average correlations for both species at all sites.

If we compare the different sites, WRF-Chem is able to capture the temporal variations best at Westmaas. This site is usually upwind of local, urban sources and the statistics mainly apply to background conditions. In

contrast, Zweth is highly affected by local sources in the Rijnmond area. Due to the spatiotemporal variability in these emissions, the daily average concentrations show large day-to-day variability that is not fully reproduced by WRF-Chem which results in lower correlations. The other two sites are situated in the domain with 4x4km resolution. Their correlation is generally slightly lower than for Westmaas, likely because WRF-Chem misses some smaller spatial variations in the concentration fields and meteorological conditions at this resolution. The bias at Cabauw is remarkably high compared to the others. We were unable to find a satisfactory explanation for

#### 3.2 Urban plume detection

this.

Now we have seen that the model performance at the mesoscale at all observation sites is satisfactory, we continue with the urban plumes. To use our framework for a future inverse modelling study and estimate the Rijnmond fluxes, we first need to consider the ability of the measurement sites to detect those urban plumes. At Lutjewad, the Rijnmond plume concentrations range from 0-4.8 ppm CO<sub>2</sub> (average of 0.3 ppm) and 0-14 ppb CO (average of <1 ppb) at a wind direction of 210°. Although sometimes detectable, the signals are hardly distinguishable from noise. Due to the large distance between source area and observation site there are only few occasions when the plume is detectable at Lutjewad and what we observe is the integrated effect of the total source area. Besides the increased dilution, the Rijnmond emissions will be mixed with other emissions and plumes from other nearby sources will dominate the observed signal. In contrast, if we select the wind sector with the port as source area, the maximum CO<sub>2</sub> and CO signals at the urban Zweth site are 29.9 ppm and 220

ppb, respectively. The average signals are 9.8 ppm  $CO_2$  and 34 ppb CO, which are large enough to be visible over the diurnal variations. However, there are large differences between wind directions, as the Zweth site will

- be affected by different parts of the source area. Therefore, the variability of the signal is large. Cabauw is located in between Lutjewad and Zweth and shows large signals, although less variable as at Zweth. Since = Lutjewad is not often affected by the urban plume from Rijnmond and is dominated by other signals, we only consider Zweth and Cabauw in the further analyses and use Westmaas for the background.
- The differences in the magnitude and variability of the signal between the measurement sites, and therefore the 310 amount of information that the plumes contain, has consequences for the use of those sites in an inversion. We illustrate this by comparing the WRF-Chem concentration ratios of CO and CO<sub>2</sub> in the urban plume to the ratio in the emissions used to drive WRF-Chem at Zweth and Cabauw. The concentration ratio is denoted  $\Delta CO:\Delta CO_2$ , where  $\Delta$  indicates that we look at a concentration increment over a background, namely only the plume concentration. The probability density functions in Fig. 5 illustrate the likelihood that a modelled concentration
- ratio takes a certain value. The smaller the distribution, the less variable the ratios are and the more likely a ratio is to take the mean value (largest probability). The mean emission ratio of the Rijnmond area (black dashed line, 2.2 ppb ppm<sup>-1</sup>) is taken from the emission inventory, taking into account the temporal profiles of the separate emission categories. Although one value is given for the emission ratio, in reality this value also has a range of several ppb ppm<sup>-1</sup> depending on the time of day, although not as large as the modelled distributions.
- We see that the mean of the distribution at Zweth is further from the emission ratio than is the mean of Cabauw. Also, its distribution is much wider. Since the Zweth measurement site is affected by different source areas with distinct emission ratios depending on the wind direction, the concentration ratios show a large variability (Super et al., 2016). To examine the width of the Zweth distribution in more detail, we added the distributions for two separate source areas by selecting events from specific wind sectors. Zweth-Rotterdam is
- illustrative for an urban residential area with a high modelled concentration ratio of 6.1 ppb ppm<sup>-1</sup>. The Zweth-port area contains mostly industrial and energy production sources and the concentrations have a mean modelled ratio of 3.5 ppb ppm<sup>-1</sup>). A similar distinction can be made in the observations, with a mean observed ratio of 8.6 ppb ppm<sup>-1</sup> for Zweth-Rotterdam (circle) and 3.7 ppb ppm<sup>-1</sup> for Zweth-port (star). Due to these distinct emission ratios and satisfactory model-observation agreement, we are able to separate the urban signal from the industrial
- signal with WRF-Chem.

Moreover, the mean modelled concentration ratio for Zweth-Rotterdam is reasonably close to the emission ratio of 6.0 ppb ppm<sup>-1</sup> for the entire residential area. This high ratio is due to the dominance of road traffic, which has an emission ratio of about 16 ppb ppm<sup>-1</sup> according to the emission inventory for the Rijnmond area. In contrast, the modelled concentration ratio for Zweth-port is much higher than the reported emission ratios of 1.1

- 335 ppb ppm<sup>-1</sup>. This discrepancies is likely related to the presence of many stack emissions in this area. These large emissions have a have a small emission ratio ~1 ppb ppm<sup>-1</sup> or lower for industrial processes and energy production that dominates the total emission ratio. However, stack emissions can be transported over the Zweth site and not be visible in the observations, especially for stacks in the close vicinity of Zweth. Therefore, the modelled concentration ratio can turn out much higher than what is expected based on the emission inventory
- including stack emissions. Indeed, the emission ratio of the entire Rijnmond area without point sources would even be 5.5 ppb ppm<sup>-1</sup>. For Cabauw this discrepancy is less pronounced, because the high stack emissions are already better mixed with the other emissions and the mean ratio is closer to the emission ratio including stack emissions. We therefore argue that Cabauw would be a better site for an overall top-down inversion study than Zweth due to its integrating power. The Zweth site can be used to separate between source areas and source

types by looking at different wind sectors and concentration ratios. Therefore, such a site could be very useful to get more details about the distribution of the emissions throughout the city and source apportionment.

# 3.3 Urban plumes in WRF-Chem

To examine how well WRF-Chem represents the urban signals, we compare the simulated plumes to the observations at Zweth and Cabauw. First, we apply a wind sector selection of 150-220° for Zweth and 240-270°

- for Cabauw. Then we need to determine which part of the observations can be attributed to the urban plume by subtracting a background contribution. The background contribution is determined by calculating the average diurnal cycle at Westmaas during westerly winds (marine regime), which is typical for the sampled wind sectors. That part of the CO<sub>2</sub> and CO measurements that is above the background value is considered to be the urban signal. These selection criteria are set to ensure we sample the Rijnmond plume and exclude other source areas and background influences. Note that we choose to include all times of the day rather than applying a daytime
  - selection criteria. This is done to ensure we also sample the morning and afternoon rush hours which cause large peaks in the concentrations. Zweth and Cabauw sample the Rijnmond plume about 30% and 10% of the time.

We find that WRF-Chem performs best at the Cabauw site, showing the highest  $R^2$  for both species and somewhat smaller biases (Table 4). However, note that the average plume concentrations at Zweth are about two

- 360 times larger than the concentrations at Cabauw such that the relative biases are similar. The small correlation coefficients indicate that WRF-Chem is unable to represent the timing of the urban signals sufficiently. The negative biases can be the result of too intense mixing in WRF-Chem, which results in an underestimation of the downwind concentration. If we take a closer look at the distribution of the  $CO_2$  plume concentrations ( $\Delta CO_2$ ) for the selected hours (Fig. 6), we find that the width of the observed distribution at Zweth is not well represented by
- WRF-Chem. This means that part of the observed variability is not captured by the model. At Cabauw, the observed distribution is much narrower and the WRF-Chem distribution is very similar to the observed one (not shown). This indicates that WRF-Chem is suitable to simulate concentrations at this regional site.

#### 3.4 Urban plumes in OPS

Now we have seen that WRF-Chem lacks detail to describe the urban plumes close to the source area, we will examine the use of the OPS plume model. To compare the OPS and WRF-Chem representations of the urban plume we select the same signals as in the previous section. The relative biases at Zweth are reduced for both species (see Table 4) when using OPS\_wm (with WRF meteorology), while OPS\_om (with observed, interpolated meteorology) changes the bias of CO<sub>2</sub> from a model underestimation to a model overestimation. However, OPS\_om shows the largest R<sup>2</sup> for both CO and CO<sub>2</sub> with explained variance up to twice that of