# Peer review of "A multi-model approach to monitor emissions of CO2 and CO in an urban-industrial complex"

_Atmospheric Chemistry and Physics, 2016_

## Referee Comment (RC1) · Anonymous Referee #1 · 20 Sep 2016

The paper by Super et al. seeks to interpret observations of CO2 and CO with a combination of Eulerian and Lagrangian models and attempts to make conclusions about the modeling framework, as well as the observational network.

However, I found the paper to be difficult to follow in what it seeks to accomplish, and I often do not see robust evidence for the claims it makes. Therefore, I cannot recommend publication.

MAJOR POINTS: (1) The concept of a "background" is mentioned throughout the paper, but the concept remains nebulous without adequate clarification. Westmaas is referred to as a "background" site while both Cabauw and Lutjewad are referred to as "regional background" sites. What are the differences between these different categories? Similarly, towards the end of the paper the terminology of "representative background" versus "regional background" stations are introduced. What do these mean? Can the authors be more quantitative by referring to spatial lengthscales relative to urban lengthscales and backing them up with observations and models? Furthermore, exactly how the background is accounted for within the analyses is also confusing. It appears that sometimes the background comes from global model products (e.g., CarbonTracker) while other times the background is determined from the observations (e.g., average diurnal cycle at Westmaas during westerly winds). What are the pros and cons of using a model-derived boundary condition versus one derived from an observed time series? These approaches are taken without motivation while I believe accounting for the background is a critical part of designing an observational network to isolate and quantify the urban signal. For example, what if the Cabauw observations are used for background determination? How much additional error would this incur? These issues remain unexplored. I also do not see the evidence supporting the claims of "Cabauw is a suitable regional background site" and "Westmaas provides reasonable background constraints to determine the urban plume signal."

(2) For a paper that makes strong claims about the value of the OPS Gaussian plume model I find the paucity of technical details about OPS to be a significant weakness. For instance, what happens to the Gaussian plumes as they are transported far away from the sources? Do they undergo "puff splitting" or "puff merging" as some other models do? Where do the turbulence variables to drive the Gaussian plumes come from? How does OPS make use of observed meteorology? The met variables are observed at point scale; how are they interpolated in space? And exactly where are the meteorological observational sites used in these study? Another OPS detail that need to be brought up earlier is the roughness length, which was only mentioned near the end of the paper.

(3) The comparison of model performance between WRF-Chem versus OPS can be much more sophisticated. First, the weakness of WRF-Chem vis-a-vis OPS is attributed to vertical and horizontal dilution. Can most of this problem (particularly pronounced during stable conditions) can be addressed by simply suppressing the vertical dilution of surface emissions within WRF-Chem? Can the authors test this? And how much of the weaknesses in WRF-Chem was due to erroneous windfields simulated by WRF itself? The authors pick out various schemes (e.g., YSU PBL scheme) as mentioned in Sect. 2.1.3, but how were these selected in the first place? Have the authors compared the simulated windfields against observed windfields?

(4) How the authors determined the "urban plume" mentioned in the first paragraph of Sect. 3.2 was entirely unclear to me. Were the plume strengths based on observations or the model? If the former, then how was the background determined (which relates back to issue (1) above)? On somewhat similar note, I had difficulties later on in Sect. 3.2 regarding the $CO:CO_2$ ratios. If $CO:CO_2$ ratios were observed, did they come from a regression of absolute CO and $CO_2$ concentrations? Or were the backgrounds subtracted out? If so, what was used for the background? And why weren't the observed distributions of $CO:CO_2$ ratios shown in Fig. 5 (only the means were shown)?

(5) The value of Cabauw is difficult to ascertain for me. The model suffers from the largest biases for daily concentrations at Cabauw. Later on in the paper the authors also claim that "WRF-Chem performs best at the Cabauw site" for simulating urban plumes. But wouldn't the biases affect modeling of urban plumes? How these two contrasting points are reconciled is unclear to me. Also see point (6) below regarding the potential problem during the growing season.

(6) The authors chose the months of Oct~Dec to carry out their study. The biospheric photosynthetic signal is much weaker during this time. How would their conclusions regarding the observational network change if months during the growing season are selected? Wouldn't this cause problems at a more removed site like Cabauw?

OTHER SPECIFIC POINTS Line 254~255: I believe that the local contribution also depends on wind direction and not just the regional contribution. Doesn't it matter which part of the urban area a site is sampling? Similarly, I disagree that advection

solely affects the background CO2 mole fraction, as mentioned in Line 130. If one writes out the tracer-transport equation, the advection term shows up prominently.

Table 3: What happens if you subset the time period to afternoon only? Many studies focus only on the afternoon due to difficulties in modeling nighttime mixing.

Fig. 1: The observation sites are difficult to pick out, and the names of the sites should be added to the figure. Lat/lon should also be added to the figure. Another helpful addition would be to overlay the sites onto a map of CO2 or CO emissions from the inventory to help the reader assess the locations of the sites relative to anthropogenic sources.

Fig. 3: How are "ff regional" and "ff local" defined? Should explain in the main text. Also, I suggest using a less prominent color for the background. Perhaps gray instead of the current yellow, which I find very distracting.

Fig. 5: Why are the observed distributions of CO:CO2 ratios not shown?

---

## Referee Comment (RC2) · Anonymous Referee #2 · 28 Nov 2016

**General**

The study "A multi-model approach to monitor emissions of CO2 and CO in an urban-industrial complex" by Ingrid Super and co-workers investigates the possibility to estimate anthropogenic $CO_2$ emissions from an urban complex by a combination of atmospheric observations and transport models. Two different model types and three different simulations of $CO_2$ and $CO$ are used to demonstrate the ability of the model system to correctly reproduce observed concentrations and their ratios. The study draws some valuable conclusions on the kind of observing system required to monitor city-scale anthropogenic emissions. The work is generally presented well and merits publication after a number of rather minor issues (mostly clarifications and additional discussion and justification) as outlined below have been addressed.

**Major comments**

**Performance of WRF:** The manuscript would benefit from a more detailed discussion on the performance of the high-resolution WRF simulations in terms of meteorological variables. For the OPS simulations two sets of simulations (observation-based and WRF-based) are presented to allow for a fair comparison between OPS and WRF concentrations. The results then show that the observation-based OPS outperforms the WRF-based OPS simulations. Part of this is obviously due to the deficiencies of the WRF simulated meteorological conditions. Therefore, it should be shown how WRF performs in terms of wind sped and direction, atmospheric stability and atmospheric boundary layer height. To this purpose a comparison between the meteorological parameters driving OPS would suffice. This subject is briefly touched on in the Discussion section, but it should be given more room in the results as well.

**Minor comments**

**Plume model type:** The manuscript introduces the plume model (OPS) as a Lagrangian model (L15), whereas later on it becomes clear that OPS is a Gaussian plume model. Although Gaussian plume models are often categorised as Lagrangian, I would prefer if the term Gaussian would be used throughout the text in order to distinguish the present model from Lagrangian particle dispersion models. See also L64.

**L28:** Please define 'extreme $CO_2$ pollution events' here.

**L66f:** Replace with: 'However, Eulerian models assume that chemical species are instantly mixed within individual grid boxes, which may enhance ...'

**L70f:** Be more specific. In which situations and on which scales can a plume model improve the simulated concentrations field. Why is this the case? A plume model does not have to be computationally expensive. Actually the main reason for their use is their simplicity and their low computational costs. The question of computational time should rather be discussed from the perspective of a Eulerian model with sufficient grid resolution to resolve the scale that is targeted by the plume model. In that case the plume model most certainly will be the computationally cheaper solution.

**L75:** It is somewhat confusing to speak of 'three modelling frameworks' and then only two models are listed. Please clarify that one of them is used with two different meteorological data sets.

**L77ff:** The discussion on other studies that have used high-resolution modelling and emission inventories to simulate greenhouse gas emissions on the urban scale should be extended. As correctly stated Oney et al. (2015) did not focus on urban areas specifically, but there are other studies that have used different model frameworks to tackle urban scale greenhouse gas emissions, some of them are already mentioned later on in the manuscript. However, it would be good to put the present study into context by referencing previous work. For example: McKain et al. (2015); Feng et al. (2016); Boon et al. (2016); Wu et al.

(2016); Bréon et al. (2015); Brioude et al. (2013)

**L88:** Instead of contrasting 'urban' with 'industrial' it may be better to contrast 'residential and transport' with 'industrial'. 'Urban' is a bit vague and may include 'industrial' sources as well.

$CO_2$ **budget, L112ff:** Why is the stratospheric component discussed separate from the background? This seems a bit arbitrary since other $CO_2$ sources (e.g., biomass burning) are also not explicitly mentioned in equation 1 and are assumed to be part of the background. Would CarbonTracker not cover strat-trop exchange as well? In a way it would be better to first mention how the background is derived and which $CO_2$ contributions can be expected to be presented by the background and then discuss the regional contributions. What is also missing from the equation is the use of bio-fuels. If these are supposed to be accounted for by the fossil fuel term then the latter should be renamed to 'anthropogenic' instead.

$CO$ **budget, L133:** The assumption that hydrocarbon oxidation can be neglected as a source of CO should be discussed in more detail, since there are several studies that showed that the 'secondary' CO contribution should not be neglected (Griffin et al., 2007) or is even dominating (Hudman et al., 2008) for summer-time conditions in the eastern US. Please discuss these findings and why your assumption may still hold for the present study. What are the consequences of neglecting the secondary source with respect to the results of the present study? Again, there is no CO source from bio-fuel burning considered in the CO budget (see previous comment). The given CO lifetime is representative for winter-time conditions. Please point out this fact and what it implies for the present study.

**L146:** What was the vertical grid resolution in the lowest 300 m of the model? This is important for the interpretation of the model-observation comparison. Would the model be able to resolve shallow stable night-time boundary layers? It would also be interesting to know how model concentrations were extracted for the location of observations (nearest grid box, horizontal and vertical interpolation, ...)?

**L150:** What kind of advection scheme was used in WRF? Was it locally mass conserving? This is important in terms of the analysis of $CO/CO_2$ ratios, which are not necessarily conserved by every advection scheme. What is the impact of using the urban module in WRF? Does it improve the model's performance in the study area?

**equation 4 and 5:** The units and/or magnitudes of SW and the factor on R, as explained in the text, must be wrong. SW and the factor on R should be unitless and yield unity if integrated over a whole month.

**equation 4:** The sub-monthly scaling for photosynthetic uptake only depends on available solar radiation and neglects any short term effects of water stress, temperature and $CO_2$ concentration on photosynthesis. Can this be justified in the context of the study?

**section 2.2:** Spend more time on the limitations of the OPS model approach. Basically the plume model gives a steady state solution under given meteorological conditions whether or not these really persisted for an hour. The model probably also assumes horizontal homogeneity. Is this justified? Therefore, it is important to know up to which transport distances (and times) the model was considered. What kind of horizontal wind speed was used in the model (10 m from observations)? Does the model consider a vertical gradient in the horizontal wind speed? Gaussian plume models are known to be invalid under low wind speed conditions. What was done for horizontal wind speeds $< 1$ m/s?

**L188:** Please clarify what is included in 'local' emissions. Are these the same as 'the Rijnmond area' emissions, which were treated separately in WRF and from the area indicated by the blue line in Fig 2?

**L203:** Use 'photosynthetic uptake' instead of 'biogenic' and refer to equation 2 and 4 for clarification.

**section 2.3:** It would be interesting to get some numbers on total population and emissions in the area, which would allow putting this study into perspective.

**L211 and elsewhere:**  Are the inlet heights given in m above ground or sea level?

**L255:**  Is 'the Rijnmond area' the same as indicated by the blue line in Fig 2? Please clarify.

**L267 and Fig 4:**  Please indicate the period of easterly winds as a gray shaded area behind the $CO_2$ and CO time series.

**L271f:**  Obviously the bias in $CO_2$ will be small compared to its atmospheric abundance. Discuss some of the comparison statistics as given in Table 3. It would also be good to contrast these daily statistics with hourly statistics in order to see how well the diurnal evolution can be captured.

**L276f:**  What could be the reason of the reduced performance during easterly advection? Is this more likely related to transport errors or to underestimated emissions? CO emissions used in TNO-MACC are known to be too low for most of central and eastern Europe. Could this be the main reason for the underestimation?

**L287:**  Add another reference to Table 3.

**L289f:**  Did the bias for $CO_2$ at Cabauw exist at all elevations?

**L296:**  Which simulation is referred to in this sentence?

**L298:**  Noise as in observational noise or variability from other sources?

**L314f and Fig 5:**  It is not clear how this pdf was derived. Is it the variability over time or somehow accessed from transport uncertainties? Why would it follow a Gaussian distribution? Why was the same kind of pdf not given for the observations as well? Please add the empirical pdf for Zweth-port and Zweth-Rotterdam and not just the mean values (maybe as bars). Give the emission ratios for the port and Rotterdam area (as described in the text) also as vertical lines.

**L328f:**  I don't understand this conclusion. You can distinguish those sources in the WRF simulations anyway, since you are using tagged tracers for different regions. Wouldn't it make more sense to conclude that you can distinguish between urban and industrial in the measurements?

**L343f:**  However, this conclusion relies on the applied model system. Using a local scale dispersion model that considers individual stack emissions and plume rise, one should be able to use even the Zweth data in an atmospheric inversion.

**L465f:**  This should also be related to the limitations of the plume model as to be discussed in more detail in the methods section.

**L467:**  Be more concrete about the distances here. Due to its assumptions of horizontal homogeneity a Gaussian plume model should not be applied at distances larger 50 km anyway.

**L491:**  This should be discussed in more detail in the results section. See major comment above.

**L501f:**  The possibility to combine the high-resolution WRF simulations also with Lagrangian particle dispersion simulations in the receptor mode (e.g. FLEXPART or STILIT) should be discussed here as well.

**L505:**  The distinction between the two background sites has to be clarified. One represents a larger scale background mostly unaffected by emissions from the region of interest, whereas the second represents the emissions from the region of interest at a distance that allows for sufficient mixing similar to that in the applied transport model.

**Fig 1:**  Please include longitudes and latitude labels on the axis.

**Fig 2:**  Please include a scale indicator and/or longitudes and latitudes.

**Fig 3:**  What is the time base of the shown data (daily/hourly)? Please add information to caption.

**Fig 6:**  Somehow it looks as if the integral over the different pdfs does not yield 1 as it should. For example the red curve is almost always larger than the blue curve. How is that possible? Please check. Are negative values for $\Delta CO_2$ considered?

**Technical comments**

**L22:**  Add 'model' in front of 'framework'.

**L69:**  Replace with: 'The magnitude of the corresponding error depends on ...'

**L75:**  Replace 'Therefore' by 'Here'.

**L306:**  There is an unintended '=' in the text.

**References**

Boon, A., Broquet, G., Clifford, D. J., Chevallier, F., Butterfield, D. M., Pison, I., Ramonet, M., Paris, J. D., and Ciais, P.: Analysis of the potential of near-ground measurements of CO2 and CH4 in London, UK, for the monitoring of city-scale emissions using an atmospheric transport model, Atmospheric Chemistry and Physics, 16, 6735–6756, doi:10.5194/acp-16-6735-2016, 2016.

Brioude, J., Angevine, W. M., Ahmadov, R., Kim, S. W., Evan, S., McKeen, S. A., Hsie, E. Y., Frost, G. J., Neuman, J. A., Pollack, I. B., Peischl, J., Ryerson, T. B., Holloway, J., Brown, S. S., Nowak, J. B., Roberts, J. M., Wofsy, S. C., Santoni, G. W., Oda, T., and Trainer, M.: Top-down estimate of surface flux in the Los Angeles Basin using a mesoscale inverse modeling technique: assessing anthropogenic emissions of CO, NOx and CO2 and their impacts, Atmospheric Chemistry and Physics, 13, 3661–3677, doi:10.5194/acp-13-3661-2013, 2013.

Bréon, F. M., Broquet, G., Puygrenier, V., Chevallier, F., Xueref-Remy, I., Ramonet, M., Dieudonné, E., Lopez, M., Schmidt, M., Perrussel, O., and Ciais, P.: An attempt at estimating Paris area $CO_2$ emissions from atmospheric concentration measurements, Atmospheric Chemistry and Physics, 15, 1707–1724, doi:10.5194/acp-15-1707-2015, 2015.

Feng, S., Lauvaux, T., Newman, S., Rao, P., Ahmadov, R., Deng, A. J., Diaz-Isaac, L. I., Duren, R. M., Fischer, M. L., Gerbig, C., Gurney, K. R., Huang, J. H., Jeong, S., Li, Z. J., Miller, C. E., O'Keeffe, D., Patarasuk, R., Sander, S. P., Song, Y., Wong, K. W., and Yung, Y. L.: Los Angeles megacity: a high-resolution land-atmosphere modelling system for urban CO2 emissions, Atmospheric Chemistry and Physics, 16, 9019–9045, doi:10.5194/acp-16-9019-2016, 2016.

Griffin, R. J., Chen, J., Carmody, K., Vutukuru, S., and Dabdub, D. C. D. S.: Contribution of gas phase oxidation of volatile organic compounds to atmospheric carbon monoxide levels in two areas of the United States, Journal of Geophysical Research: Atmospheres, 112, n/a–n/a, doi:10.1029/2006JD007602, 2007.

Hudman, R. C., Murray, L. T., Jacob, D. J., Millet, D. B., Turquety, S., Wu, S., Blake, D. R., Goldstein, A. H., Holloway, J., and Sachse, G. W. C. L.: Biogenic versus anthropogenic sources of CO in the United States, Geophysical Research Letters, 35, n/a–n/a, doi:10.1029/2007GL032393, 2008.

McKain, K., Down, A., Raciti, S. M., Budney, J., Hutyra, L. R., Floerchinger, C., Herndon, S. C., Nehrkorn, T., Zahniser, M. S., Jackson, R. B., Phillips, N., and Wofsy, S. C.: Methane emissions from natural gas infrastructure and use in the urban region of Boston, Massachusetts, Proceedings of the National Academy of Sciences, 112, 1941–1946, doi:10.1073/pnas.1416261112, 2015.

Oney, B., Henne, S., Gruber, N., Leuenberger, M., Bamberger, I., Eugster, W., and Brunner, D.: The CarboCount CH sites: characterization of a dense greenhouse gas observation network, Atmospheric Chemistry and Physics, 15, 11 147–11 164, doi:10.5194/acp-15-11147-2015, 2015.

Wu, L., Broquet, G., Ciais, P., Bellassen, V., Vogel, F., Chevallier, F., Xueref-Remy, I., and Wang, Y.: What would dense atmospheric observation networks bring to the quantification of city CO2 emissions?, Atmospheric Chemistry and Physics, 16, 7743–7771, doi:10.5194/acp-16-7743-2016, 2016.

---

## Author Comment (AC1) · 22 Dec 2016

We would like to thank the reviewers for their interest in our study and for the comments on our work. The review comments have been very helpful in reflecting on our own work and pointing out parts that required further improvements.

In general, our revisions aim to clarify the methodology, which both reviewers found too brief and thereby confusing at some points. Also, we have improved the methodology specifically for treatment of the "background" concentrations as suggested. Finally, we have rewritten the results sections to more clearly identify the strengths of the Gaussian plume model in contrast to the clearly identified weaknesses of the Eulerian approach.

Below we address specific issues mentioned by the reviewers point by point. The results and discussion have been updated accordingly.

**Reviewer #1**
*The paper by Super et al. seeks to interpret observations of CO2 and CO with a combination of Eulerian and Lagrangian models and attempts to make conclusions about the modeling framework, as well as the observational network. However, I found the paper to be difficult to follow in what it seeks to accomplish, and I often do not see robust evidence for the claims it makes. Therefore, I cannot recommend publication.*

*MAJOR POINTS*
*The concept of a "background" is mentioned throughout the paper, but the concept remains nebulous without adequate clarification. Westmaas is referred to as a "background" site while both Cabauw and Lutjewad are referred to as "regional background" sites.*
*1) What are the differences between these different categories?*
*Similarly, towards the end of the paper the terminology of "representative background" versus "regional background" stations are introduced.*
*2) What do these mean?*

We agree that the use of background was a weakness in the manuscript, and the different ways we referred to it did not help. Therefore, we have made a major improvement by using a single, previously described, and well-described algorithm (see Sect. 2.4). This has given us the chance to also update the terminology and remove any ambiguity on the removal of the background signal. Also, we renamed the measurement sites based on their location relative to the urban area (urban, semi-urban, rural) to avoid any confusion with the background signal.

*3) Can the authors be more quantitative by referring to spatial lengthscales relative to urban lengthscales and backing them up with observations and models?*

We are not sure what the reviewer requests from us here, but it sounds like he/she would like an observation-based definition of urban lengths scales. We are sorry, but we would not know how to address this request. We do however present an analyses of the differences between the two modeling approaches to come to the conclusion that the important differences occur at scales below about 6 km. We hope this satisfies the reviewer's request for a quantification.

*Furthermore, exactly how the background is accounted for within the analyses is also confusing. It appears that sometimes the background comes from global model products (e.g., CarbonTracker) while other times the background is determined from the observations (e.g., average diurnal cycle at Westmaas during westerly winds).*
*4) What are the pros and cons of using a model-derived boundary condition versus one derived from an observed time series?*

We understand the confusion, which was mostly the result of the terminology which we have improved. However, there is a significant difference between the background signal that we account for in the time-series analyses and the large-scale background tracer in WRF-Chem. The latter represents the lateral inflow at the domain boundaries and is indeed model-based (from CarbonTracker). In contrast, the "background" for a specific site is derived from the observed or modelled concentrations. The new method we apply for this is now clearly documented in the Methods section (Sect. 2.4).

*These approaches are taken without motivation while I believe accounting for the background is a critical part of designing an observational network to isolate and quantify the urban signal.*
*5) For example, what if the Cabauw observations are used for background determination? How much additional error would this incur?*

We understand the suggestion given by the reviewer, but we would like to point out that Cabauw is not a good large-scale characterisation site. We show in the manuscript that it is often downwind of the urban area. Therefore, using Cabauw to determine the background for all observation sites would introduce a significant error, which is likely on the order of several ppm (see for example Tolk et al. (2009) for a quantification of the fossil fuel contribution at the Cabauw site).

*These issues remain unexplored. I also do not see the evidence supporting the claims of "Cabauw is a suitable regional background site" and "Westmaas provides reasonable background constraints to determine the urban plume signal."*

An additional section has been added to the methods, explaining the concept of background and how it is used in the paper. Also, we used the background definition more consistently throughout the paper and terminology has been updated. Moreover, we renamed the measurement sites based on their location relative to the urban area (urban, semi-urban, rural) to avoid further confusion. All conclusions about the sites are now better supported by the observations.

*For a paper that makes strong claims about the value of the OPS Gaussian plume model I find the paucity of technical details about OPS to be a significant weakness.*

We apologize for this shortcoming. We have added some additional information on the OPS model in the revised manuscript (Sect. 2.2), but we would like to refer to the many papers about the model and the technical manual for more details. We believe that the Gaussian model concept is likely well-known to the reader, and the details are only of interest for those readers that have a special interest in this family of models. They will be able to find all information in the references provided.

*6) For instance, what happens to the Gaussian plumes as they are transported far away from the sources?*
*7) Do they undergo "puff splitting" or "puff merging" as some other models do?*

For each individual plume a trajectory is calculated over a period of maximum four days. The trajectory uses hourly updated meteorological conditions. For each individual plume the model determines whether it affects the receptor. If so, the Gaussian plume formulation is used to calculate the concentration caused by that particular source at the receptor point, using the true travel distance. Question 7 is not relevant to our study, as the OPS model is run in offline mode and plumes are not merged with the WRF-Chem model. Only after the simulation the tracers are added to the WRF-Chem large-scale background.

*8) Where do the turbulence variables to drive the Gaussian plumes come from?*

Turbulence is not resolved in the OPS model and the dispersion is determined by the dispersion coefficients $\sigma_y$ and $\sigma_z$. These coefficients are a function of atmospheric stability, friction velocity, Obukhov length and boundary layer height. The exact formulations can be found in the user manual referred to in Sect. 2.2 (also see http://www.rivm.nl/media/ops/v4.5.0/OPS-model-v4.5.0.pdf).

*9) How does OPS make use of observed meteorology?*
*10) The met variables are observed at point scale; how are they interpolated in space?*
*11) And exactly where are the meteorological observational sites used in these study?*

Observed wind speed, wind direction, temperature, humidity, and several other primary variables are observed at 19 locations in the Netherlands by the Royal Dutch Meteorological Office (KNMI). These variables are used to calculate secondary variables, such as the Obukhov length and sensible heat flux. The observations are interpolated over the Netherlands to a 10x10km$^2$ grid. Each station gets a weighing factor, depending on the distance between the station and a specific grid point. Afterwards, the parameters are averaged over one of the six specified regions. The trajectory location determines which region is used to describe the meteorological conditions. All this information can also be found in the user manual. Also see the figure below for the locations.

[Figure]

*Figure 2.1.*    *OPS meteorological districts (on a 10 x 10 km² grid) and location of KNMI stations.*

*Another OPS detail that need to be brought up earlier is the roughness length, which was only mentioned near the end of the paper.*

The roughness length is now mentioned in the methods section (lines 274-277).

*The comparison of model performance between WRF-Chem versus OPS can be much more sophisticated.*

We have rewritten Section 3.3 (about the comparison of WRF-Chem and OPS) using a systematic approach that describes the main differences. We hope this satisfies the reviewer.

*First, the weakness of WRF-Chem vis-a-vis OPS is attributed to vertical and horizontal dilution.*
*12) Can most of this problem (particularly pronounced during stable conditions) can be addressed by simply suppressing the vertical dilution of surface emissions within WRF-Chem?*
*13) Can the authors test this?*

The difference between OPS and WRF-Chem can be explained by several factors, including the difference in mixing. However, suppressing mixing in WRF-Chem is not easy, as it is an internal part of the dynamics. Therefore, we have performed an additional OPS simulation which uses the WRF-Chem representation of point sources (a simplified vertical distribution and a 1x1km² horizontal resolution) and WRF-Chem meteorology. Results from this run have been described in lines 425-432 and show the impact of instant mixing in WRF-Chem and the lack of point sources on average have a similar impact. Therefore, decreasing dilution by adding additional vertical levels does not solve all issues.

*14) And how much of the weaknesses in WRF-Chem was due to erroneous windfields simulated by WRF itself?*

The problem with the WRF-Chem wind direction was already highlighted in the manuscript. To emphasize this important finding, we have added some additional information about the limitations of WRF-Chem related to errors in meteorological variables (lines 361-368).

*15) The authors pick out various schemes (e.g., YSU PBL scheme) as mentioned in Sect. 2.1.3, but how were these selected in the first place?*

Our selection is based on previous WRF modelling studies in the Netherlands (lines 159-160). The referenced studies have tested the performance of this framework against meteorological and trace gas measurements at various sites.

*16) Have the authors compared the simulated windfields against observed windfields?*

Yes, as mentioned in the Discussion (lines 528-530) the bias was very large. We have now mentioned some of these findings already in the results to assist in interpreting the results (Sect. 3.2).

*How the authors determined the "urban plume" mentioned in the first paragraph of Sect. 3.2 was entirely unclear to me.*
*17) Were the plume strengths based on observations or the model?*
*18) If the former, then how was the background determined (which relates back to issue (1) above)?*
*On somewhat similar note, I had difficulties later on in Sect. 3.2 regarding the CO:CO2 ratios.*
*19) If CO:CO2 ratios were observed, did they come from a regression of absolute CO and CO2 concentrations?*
*20) Or were the backgrounds subtracted out?*
*21) If so, what was used for the background?*
*22) And why weren't the observed distributions of CO:CO2 ratios shown in Fig. 5 (only the means were shown)?*

We realize that the description led to confusion and we have updated it in a new section in the Methods (Sect. 2.4) and following our previous response to issue 1. The same method is now applied to all analyses. Figure 5 (now Fig. 4) is now updated using observations as suggested under question 22.

*The value of Cabauw is difficult to ascertain for me. The model suffers from the largest biases for daily concentrations at Cabauw. Later on in the paper the authors also claim that "WRF-Chem performs best at the Cabauw site" for simulating urban plumes.*
*23) But wouldn't the biases affect modeling of urban plumes?*
*How these two contrasting points are reconciled is unclear to me. Also see point (6) below regarding the potential problem during the growing season.*

We have updated Table 3 (as the reviewer suggested under question 27). The updated Table 3 shows that the statistics at Cabauw is very similar to that at the other sites. The bias for $CO_2$ is slightly larger than expected, but we have found out that there was a calibration problem for $CO_2$ at Cabauw which could have caused this bias. Yet, we believe that this additional bias (which is rather small for the daytime we have now selected throughout our manuscript) will not change our findings, which are mostly based on the variability in the observed signals. The statement that WRF-Chem performs best at the Cabauw site has been removed.

*The authors chose the months of Oct-Dec to carry out their study. The biospheric photosynthetic signal is much weaker during this time.*
*24) How would their conclusions regarding the observational network change if months during the growing season are selected?*
*25) Wouldn't this cause problems at a more removed site like Cabauw?*

The reviewer raises an important point about the chosen time period, as for example the validity of the $CO:CO_2$ ratio method must be strongly questioned during the summer months as suggested by Turnbull et al. (2006). This stems from both a large contribution of hydrocarbon oxidation to CO production, and from a large dominance of photosynthesis and respiration on $CO_2$. As a result, during the growing season we would expect to find smaller peak values of $CO_2$ during the day, and larger peak values of CO (assuming an equal PBL height). The biogenic could (in the absence of urban carbon sinks) be mostly accounted for in the background, which would be lower than during winter.

Due to the difficulty of exactly quantifying the biogenic fluxes, we picked these three winter months in which the urban plume emissions are not easily taken up by vegetation. Indeed, we only find a small gradient in the $CO_2$ increase due to respiration between Zweth and Cabauw and no gradient in the $CO_2$ decrease due to photosynthesis.

*OTHER SPECIFIC POINTS*
*Line 254-255*
*I believe that the local contribution also depends on wind direction and not just the regional contribution.*
*26) Doesn't it matter which part of the urban area a site is sampling?*

*Similarly, I disagree that advection solely affects the background CO2 mole fraction, as mentioned in Line 130. If one writes out the tracer-transport equation, the advection term shows up prominently.*

The reviewer is correct. We did not intend to state that only the regional contribution is dependent on advection, but just that this is an important factor for this contribution.

The model background concentration is determined by the initial and boundary conditions only and is therefore solely dependent on advection. But that indeed does not mean that e.g. the fossil fuel plumes are not a function of advection. We tried to clarify this in lines 134-135.

*Table 3*
*27) What happens if you subset the time period to afternoon only?*
*Many studies focus only on the afternoon due to difficulties in modeling nighttime mixing.*

Following this suggestion, the table has been updated using only daytime data. We have deliberately chosen to use all daytime hours (not just the afternoon), since we would like to sample also the morning rush hour, which would be excluded sampling only the afternoon hours. We found that sampling only afternoon hours, however, results in similar or slightly lower correlations and slightly more positive biases.

*Fig. 1*
*The observation sites are difficult to pick out, and the names of the sites should be added to the figure. Lat/lon should also be added to the figure. Another helpful addition would be to overlay the sites onto a map of CO2 or CO emissions from the inventory to help the reader assess the locations of the sites relative to anthropogenic sources.*

Figures 1 and 2 have been updated.

*Fig. 3*
*28) How are "ff regional" and "ff local" defined?*
*Should explain in the main text. Also, I suggest using a less prominent color for the background. Perhaps gray instead of the current yellow, which I find very distracting.*

This figure is deleted.

*Fig. 5*
*29) Why are the observed distributions of CO:CO2 ratios not shown?*

We agree that this would be helpful. In the new manuscript this figure is now purely based on observations.

**Reviewer #2**
*The study "A multi-model approach to monitor emissions of CO2 and CO in an urban-industrial complex" by Ingrid Super and co-workers investigates the possibility to estimate anthropogenic CO2 emissions from an urban complex by a combination of atmospheric observations and transport models. Two different model types and three different simulations of CO2 and CO are used to demonstrate the ability of the model system to correctly reproduce observed concentrations and their ratios. The study draws some valuable conclusions on the kind of observing system required to monitor city-scale anthropogenic emissions. The work is generally presented well and merits publication after a number of rather minor issues (mostly clarifications and additional discussion and justification) as outlined below have been addressed.*

We kindly thank the reviewer for this positive evaluation, and are pleased to read that our study comes to valuable conclusions.

**Performance of WRF:** *The manuscript would benefit from a more detailed discussion on the performance of the high-resolution WRF simulations in terms of meteorological variables. For the OPS simulations two sets of simulations (observation-based and WRF-based) are presented to allow for a fair comparison between OPS and WRF concentrations. The results then show that the observation-based OPS outperforms the WRF-based OPS simulations. Part of this is obviously due to the deficiencies of the WRF simulated meteorological conditions. Therefore, it should be shown how WRF performs in terms of wind speed and direction, atmospheric stability and atmospheric boundary layer height. To this purpose a comparison between the meteorological parameters*

*driving OPS would suffice. This subject is briefly touched on in the Discussion section, but it should be given more room in the results as well.*

We thank the reviewer for this helpful suggestion, and we have added the requested analysis to the manuscript (lines 361-368 and Table 5). For the OPS simulations with WRF-Chem four variables are replaced with respect to the original set-up of the model using observations: 2m temperature, humidity, wind speed and wind direction. We have added a comparison of the WRF-Chem simulation with observations for the four replaced variables. This clearly shows that the main difference is in the wind direction. The boundary layer height is taken from WRF-Chem for both OPS simulations to exclude the impact of different ABL heights. A previous study with a similar WRF-Chem set-up showed that the ABL height simulated at Cabauw is underestimated during night time, but during daytime the ABL height has the right order of magnitude (Bozhinova et al., 2014). However, since we applied the WRF-Chem boundary layer height to the OPS simulations, this cannot explain the difference between the different model simulations.

***Plume model type:*** *The manuscript introduces the plume model (OPS) as a Lagrangian model (L15), whereas later on it becomes clear that OPS is a Gaussian plume model. Although Gaussian plume models are often categorised as Lagrangian, I would prefer if the term Gaussian would be used throughout the text in order to distinguish the present model from Lagrangian particle dispersion models. See also L64.*

The term 'Lagrangian' has been replaced with 'Gaussian' throughout the text.

***L28:*** *Please define 'extreme CO2 pollution events' here.*

The abstract has been updated and this term has been replaced.

***L66f:*** *Replace with: 'However, Eulerian models assume that chemical species are instantly mixed within individual grid boxes, which may enhance ...'*

Done.

***L70f:*** *Be more specific. In which situations and on which scales can a plume model improve the simulated concentrations field. Why is this the case? A plume model does not have to be computationally expensive. Actually the main reason for their use is their simplicity and their low computational costs. The question of computational time should rather be discussed from the perspective of a Eulerian model with sufficient grid resolution to resolve the scale that is targeted by the plume model. In that case the plume model most certainly will be the computationally cheaper solution.*

The studies referenced here have used plume models to represent the transport of point and line source emissions at local or urban scales. The improvements due to implementing such plume model is thus likely due to improved representation of the source and its location. This has been clarified (lines 70-71).

We agree with the reviewer that a plume model is rather efficient when applied to point source emissions only, such as is currently done. However, a Eulerian model would still be needed to represent area source emissions and the computational time of the plume model should be added. Replacing the Eulerian model with a plume model for all source types and a large area would likely increase the computational cost compared to the Eulerian model. We clarified our statement (lines 72-73).

***L75:*** *It is somewhat confusing to speak of 'three modelling frameworks' and then only two models are listed. Please clarify that one of them is used with two different meteorological data sets.*

We thank the reviewer for this suggestion. We have adapted this sentence (lines 74-75).

***L77ff:*** *The discussion on other studies that have used high-resolution modelling and emission inventories to simulate greenhouse gas emissions on the urban scale should be extended. As correctly stated Oney et al. (2015) did not focus on urban areas specifically, but there are other studies that have used different model frameworks to tackle urban scale greenhouse gas emissions, some of them are already mentioned later on in the manuscript. However, it would be good to put the present study into context by referencing previous work. For example: McKain et al. (2015); Feng et al. (2016); Boon et al. (2016); Wu et al. (2016); Bréon et al. (2015); Brioude et al. (2013)*

This is done in lines 80-83.

*L88: Instead of contrasting 'urban' with 'industrial' it may be better to contrast 'residential and transport' with 'industrial'. 'Urban' is a bit vague and may include 'industrial' sources as well.*

Done.

*CO2 budget, L112ff: Why is the stratospheric component discussed separate from the background? This seems a bit arbitrary since other CO2 sources (e.g., biomass burning) are also not explicitly mentioned in equation 1 and are assumed to be part of the background. Would CarbonTracker not cover strat-trop exchange as well? In a way it would be better to first mention how the background is derived and which CO2 contributions can be expected to be presented by the background and then discuss the regional contributions. What is also missing from the equation is the use of bio-fuels. If these are supposed to be accounted for by the fossil fuel term then the latter should be renamed to 'anthropogenic' instead.*

We understand the reviewer's comment that the chosen components seem arbitrary. We had adopted the Eq. (1) from the paper of Bozhinova et al. (2014) where it includes the stratospheric component. We made some small alterations to the text and Eq. (1) to be more clear about what we assume is included in the background. Also, the biofuels have been included in the WRF-Chem simulations but we forgot to mention them in Eq. (1). They have now been added.

*CO budget, L133: The assumption that hydrocarbon oxidation can be neglected as a source of CO should be discussed in more detail, since there are several studies that showed that the 'secondary' CO contribution should not be neglected (Griffin et al., 2007) or is even dominating (Hudman et al., 2008) for summer-time conditions in the eastern US. Please discuss these findings and why your assumption may still hold for the present study. What are the consequences of neglecting the secondary source with respect to the results of the present study? Again, there is no CO source from bio-fuel burning considered in the CO budget (see previous comment). The given CO lifetime is representative for winter-time conditions. Please point out this fact and what it implies for the present study.*

We agree with the reviewer that oxidation processes can have a large impact on the CO concentration in the atmosphere and hence impact, for example, the validity of the $CO:CO_2$ ratio method. However, the studies this reviewer mentions were all performed during summer months with conditions favourable for photochemical oxidation of hydrocarbons. We have picked three late fall/winter months to limit the impact of hydrocarbon oxidation and photosynthetic uptake. We have added some lines to our manuscript that raise this issue and the possible consequences for our findings (lines 137-144). The biofuel source for CO has not been added, as we do not have any information about these fluxes. But given the very small contribution of biofuel combustion to the total anthropogenic $CO_2$ emissions we assume this term to be negligible for CO.

*L146: What was the vertical grid resolution in the lowest 300 m of the model? This is important for the interpretation of the model-observation comparison. Would the model be able to resolve shallow stable night-time boundary layers? It would also be interesting to know how model concentrations were extracted for the location of observations (nearest grid box, horizontal and vertical interpolation, . . . )?*

The ability of WRF-Chem to simulate the ABL height has been shown to be poor during night time (Bozhinova et al., 2014). This bias can play a role in the model-observation comparison and we try to limit this error by sampling only daytime hours. We have added some information about the model sampling and vertical resolution to Sect. 2.1.3. There are 4 vertical levels in the lowest 300m.

*L150: What kind of advection scheme was used in WRF? Was it locally mass conserving? This is important in terms of the analysis of CO/CO2 ratios, which are not necessarily conserved by every advection scheme. What is the impact of using the urban module in WRF? Does it improve the model's performance in the study area?*

The default (positive-definite) advection scheme was used, which is both locally and globally mass conserving.

We have done simulations with and without the UCM model and the impact on the time series statistics is small and inconsistent. Differences in $R^2$ are less than 0.01 and differences in biases are up to 2 ppm for $CO_2$ and 10 ppb for CO (not consistently better for either of the simulations) for the total simulation period based on hourly data.

***equation 4 and 5:*** *The units and/or magnitudes of SW and the factor on R, as explained in the text, must be wrong. SW and the factor on R should be unitless and yield unity if integrated over a whole month.*

We thank the reviewer for pointing out this mistake. We have now put the correct units in the text.

***equation 4:*** *The sub-monthly scaling for photosynthetic uptake only depends on available solar radiation and neglects any short term effects of water stress, temperature and CO2 concentration on photosynthesis. Can this be justified in the context of the study?*

The reviewer is correct that this method neglects some environmental factors that could affect the photosynthesis. However, given that we only look at several late fall/winter months in which the photosynthesis is small, we argue that the impact of those environmental factors will be negligibly small compared to the large fossil fuel fluxes (lines 193-195).

***section 2.2:*** *Spend more time on the limitations of the OPS model approach. Basically the plume model gives a steady state solution under given meteorological conditions whether or not these really persisted for an hour. The model probably also assumes horizontal homogeneity. Is this justified? Therefore, it is important to know up to which transport distances (and times) the model was considered. What kind of horizontal wind speed was used in the model (10 m from observations)? Does the model consider a vertical gradient in the horizontal wind speed? Gaussian plume models are known to be invalid under low wind speed conditions. What was done for horizontal wind speeds < 1 m/s?*

The reviewer raises an important concern about the extend at which a Gaussian plume model is applicable. Indeed, the OPS model uses several simplifications by assuming certain homogeneity in space and time and any simplification introduces an error in the modelled concentrations. We mention this in lines 220-221. Moreover, a vertical wind profile is calculated based on the Monin-Obukhov similarity theory and the observed 10 m wind speed, and the turning of the wind with height is also taken into account. Excluding low wind speed conditions does not affect our results significantly, as we sampled only daytime conditions and the amount of low wind speed hours is low in the sample.

We agree with the reviewer that a discussion on the time and spatial scales at which the OPS model could be used is very relevant. We have added our recommendations about this to the Discussion. Our findings indicate that a plume model should only be considered up to 6-10 km from the city limits. Indeed, at Cabauw (~30 km from the city centre) we find no improvement using the OPS model.

***L188:*** *Please clarify what is included in 'local' emissions. Are these the same as 'the Rijnmond area' emissions, which were treated separately in WRF and from the area indicated by the blue line in Fig 2?*

The 'local' emissions are those emissions taking place within domain 4 of WRF-Chem, which is only slightly larger than the Rijnmond area lined out in Fig. 2. We no longer refer to local fluxes in this context, but use larger Rijnmond area emissions throughout the text.

***L203:*** *Use 'photosynthetic uptake' instead of 'biogenic' and refer to equation 2 and 4 for clarification.*

Done.

***section 2.3:*** *It would be interesting to get some numbers on total population and emissions in the area, which would allow putting this study into perspective.*

Some additional information has been added to Sect. 2.3 and an emission map is now shown (Fig. 2).

***L211 and elsewhere:*** *Are the inlet heights given in m above ground or sea level?*

The heights are given above ground level, which has clarified in the manuscript.

***L255:*** *Is 'the Rijnmond area' the same as indicated by the blue line in Fig 2? Please clarify.*

The Rijnmond area mentioned here refers to the WRF-Chem domain 4 and is slightly larger than the outlined area in Fig. 2. This has been clarified.

***L267 and Fig 4:*** *Please indicate the period of easterly winds as a gray shaded area behind the CO2 and CO time series.*

We have removed any references to the differences between the period of easterly advection and the rest of the time series. The reason for this is that we include all data (including November) in the remainder of our analyses and therefore a subdivision between months seems irrelevant.

***L271f:*** *Obviously the bias in CO2 will be small compared to its atmospheric abundance. Discuss some of the comparison statistics as given in Table 3. It would also be good to contrast these daily statistics with hourly statistics in order to see how well the diurnal evolution can be captured.*

Table 3 has been updated based on a suggestion of reviewer #1. We have now only included daytime hours (8-17h LT), which are the hours included in all our analyses. Therefore, the agreement at night time is less relevant. Nevertheless, using hourly statistics would show a reduction in $R^2$ of about 0.15 at the sites close to the urban area (Zweth and Westmaas), while there is no significant change in the $R^2$ at Cabauw or Lutejwad. The effect could possibly be attributed to the large uncertainty in the night time boundary layer height, that is especially relevant at sites that are affected by local emissions.

***L276f:*** *What could be the reason of the reduced performance during easterly advection? Is this more likely related to transport errors or to underestimated emissions? CO emissions used in TNO-MACC are known to be too low for most of central and eastern Europe. Could this be the main reason for the underestimation?*

We think that the error in the emission could be part of the problem, as suggested by the reviewer. However, we also found the largest bias in the wind direction in this period (see Discussion lines 528-530). Nevertheless, as mentioned before we removed any references to the differences between the period of easterly advection and the rest of the time series.

***L287:*** *Add another reference to Table 3.*

This part has been removed.

***L289f:*** *Did the bias for CO2 at Cabauw exist at all elevations?*

Yes, the bias is found at all four levels. Recently, we have found out that there was a calibration problem at the Cabauw site for $CO_2$ which could have caused this bias. Yet, we believe that this additional bias (which is rather small for the daytime we have now selected throughout our manuscript) will not change our findings, which are mostly based on the variability in the observed signals.

***L296:*** *Which simulation is referred to in this sentence?*

We have clarified that this section is about WRF-Chem.

***L298:*** *Noise as in observational noise or variability from other sources?*

This sentence has been removed.

***L314f and Fig 5:*** *It is not clear how this pdf was derived. Is it the variability over time or somehow accessed from transport uncertainties? Why would it follow a Gaussian distribution? Why was the same kind of pdf not given for the observations as well? Please add the empirical pdf for Zweth-port and Zweth-Rotterdam and not just the mean values (maybe as bars). Give the emission ratios for the port and Rotterdam area (as described in the text) also as vertical lines.*

The PDF now contains all the observed concentration ratios in the urban plume (i.e. above the background). In fact, the fit should be estimated with a skewed distribution, since the ratios cannot take values less than zero. However, we compared the skewed and Gaussian fits and the conclusions remain the same. We therefore choose to use the Gaussian fit to be able to express the statistics in, easy to interpret, means and standard deviations.

***L328f:*** *I don't understand this conclusion. You can distinguish those sources in the WRF simulations anyway, since you are using tagged tracers for different regions. Wouldn't it make more sense to conclude that you can distinguish between urban and industrial in the measurements?*

We understand the reviewer's confusion. Fig. 5 (now Fig. 4) has therefore been updated using observations instead.

*L343f: However, this conclusion relies on the applied model system. Using a local scale dispersion model that considers individual stack emissions and plume rise, one should be able to use even the Zweth data in an atmospheric inversion.*

We agree with the reviewer that Zweth could be used in an inversion if a suitable model framework is available. However, the point that we would like to make here is that you would need data from many different wind directions to get the total emissions of the Rijnmond area and Zweth is likely to miss some stack emissions. In contrast, Cabauw gives an integral constraint on the Rijnmond emissions with measurements from only a few, dominating wind directions. A sentence has been added to stress this (lines 461-462).

*L465f: This should also be related to the limitations of the plume model as to be discussed in more detail in the methods section.*

This part has been removed.

*L467: Be more concrete about the distances here. Due to its assumptions of horizontal homogeneity a Gaussian plume model should not be applied at distances larger 50 km anyway.*

Yes, we agree with the reviewer that a Gaussian plume model has a limited spatial extent. We have added a discussion on this topic and the implications for our study (see Discussion).

*L491: This should be discussed in more detail in the results section. See major comment above.*

A more detailed comparison has been added (also see our previous comments).

*L501f: The possibility to combine the high-resolution WRF simulations also with Lagrangian particle dispersion simulations in the receptor mode (e.g. FLEXPART or STILIT) should be discussed here as well.*

We would like to thank the reviewer for this suggestion. We have added this to the Discussion (lines 543-545).

*L505: The distinction between the two background sites has to be clarified. One represents a larger scale background mostly unaffected by emissions from the region of interest, whereas the second represents the emissions from the region of interest at a distance that allows for sufficient mixing similar to that in the applied transport model.*

We understand the confusion and have updated the terminology applied to the measurement sites throughout the manuscript.

*Fig 1: Please include longitudes and latitude labels on the axis.*

Done.

*Fig 2: Please include a scale indicator and/or longitudes and latitudes.*

Done.

*Fig 3: What is the time base of the shown data (daily/hourly)? Please add information to caption.*

This figure is removed.

*Fig 6: Somehow it looks as if the integral over the different pdfs does not yield 1 as it should. For example the red curve is almost always larger than the blue curve. How is that possible? Please check. Are negative values for ΔCO2 considered?*

We thank the reviewer for pointing this out. Indeed an error was made, but the figure has now been corrected and updated. Negative values are not considered.

***L22:*** *Add 'model' in front of 'framework'.*

Done.

***L69:*** *Replace with: 'The magnitude of the corresponding error depends on ...'*

Done.

***L75:*** *Replace 'Therefore' by 'Here'.*

Done.

***L306:*** *There is an unintended '=' in the text.*

The '=' has been removed.

**References**

Bozhinova, D., Van Der Molen, M. K., Van Der Velde, I. R., Krol, M. C., Van Der Laan, S., Meijer, H. A. J., and Peters, W.: Simulating the integrated summertime $\delta14CO_2$ signature from anthropogenic emissions over Western Europe, Atmos. Chem. Phys., 14, 7273-7290, 10.5194/acp-14-7273-2014, 2014.

Tolk, L. F., Peters, W., Meesters, A. G. C. A., Groenendijk, M., Vermeulen, A. T., Steeneveld, G. J., and Dolman, A. J.: Modelling regional scale surface fluxes, meteorology and $CO_2$ mixing ratios for the Cabauw tower in the Netherlands, Biogeosciences, 6, 2265-2280, 2009.

Turnbull, J. C., Miller, J. B., Lehman, S. J., Tans, P. P., Sparks, R. J., and Southon, J.: Comparison of $14CO_2$, CO, and $SF_6$ as tracers for recently added fossil fuel $CO_2$ in the atmosphere and implications for biological $CO_2$ exchange, Geophys Res Lett, 33, 1-5, 2006.